# Syntax Matters: Towards Spoken Language Understanding via Syntax-Aware Attention

Yifeng Xie♡   Zhihong Zhu♣   Xuxin Cheng♣   Zhiqi Huang♣   Dongsheng Chen♣

♡Guangdong University of Technology,   ♣Peking University

evfxie@gmail.com

## Abstract

Spoken Language Understanding (SLU), a crucial component of task-oriented dialogue systems, has consistently garnered attention from both academic and industrial communities. Although incorporating syntactic information into models has the potential to enhance the comprehension of user utterances and yield impressive results, its application in SLU systems remains largely unexplored. In this paper, we propose a carefully designed model termed **S**yntax-**a**ware a**t**tention (SAT) to enhance SLU, where attention scopes are constrained based on relationships within the syntactic structure. Experimental results on three datasets show that our model achieves substantial improvements and excellent performance. Moreover, SAT can be integrated into other BERT-based language models to further boost their performance.

## 1 Introduction

In recent years, there has been a surge of research interest in dialogue systems from academia and industry, where SLU (Chen et al., 2017; Huang et al., 2020; Chen et al., 2022b) is a crucial component, responsible for identifying user intents based on their natural language utterances (Huang et al., 2021c, 2022; Chen et al., 2022a; Zhu et al., 2023c,b). Many works have been proposed to enhance the understanding of utterance semantics, leading to remarkable successes.

While language models (LMs) have the ability to capture contextual semantics to some extent, their performance can be further improved by integrating external knowledge (Cheng et al., 2023c). The inclusion of additional knowledge in LMs has become a growing trend, and researchers have extensively explored the integration of syntax information into pre-trained LMs (Bai et al., 2021; Zhang et al., 2020, 2022; Gong et al., 2022). Incorporating syntax information through parser dependencies into PLMs improves performance compared to

| Intent | Utterance |
|--------|-----------|
| NUM | How many **watts make** a **kilowatt** ? |
| ABBR | What is the **abbreviation** for micro ? |
| DESC | What **caused** the **death** of Bob Marley ? |
| ENTY | What **kind** of creature is a **coot** ? |
| LOC | Where **is** the **Valley of** the Kings ? |
| HUM | Who was the second **man** to **walk** on the moon ? |

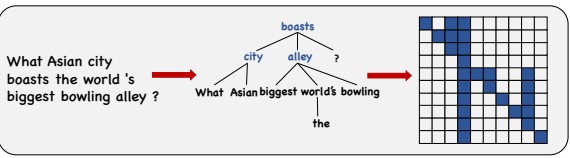

Figure 1: Intent classification examples on TREC6 dataset **(Top)** and syntax parsing process **(Bottom)**.

training with utterances alone. However, relevant research on SLU is lacking.

To address the deficiency, we propose **S**yntax-**a**ware a**t**tention (SAT) for SLU. Specifically, we first utilize a pre-trained dependency parse tree structure to generate the related nodes for each word in a sentence. These related nodes, known as syntax-aware attention mask, consist of all ancestor nodes and the word itself in the dependency parsing tree, treating the word as a child node. Examples of intent detection and syntax parsing are shown in Figure 1. To effectively incorporate **S**yntax-**a**ware **At**tention **M**odule (SATM), we also design the **D**ual **C**ontext **F**usion **M**odule (DCFM) to merge the original Transformer (Vaswani et al., 2017) with the syntax-aware Transformer. This enables SAT to provide more linguistically inspired representations for SLU.

To summarize, our contributions are three-fold: (1) SAT can capture the syntactic structure information from utterances for enhancing SLU. It can be straightforward integrated into BERT-based models. (2) In order to effectively merge syntactic information, various feature fusion methods were explored with the aim of enhancing linguistic representation. (3) Experiments demonstrate the effectiveness on three SLU benchmarks.

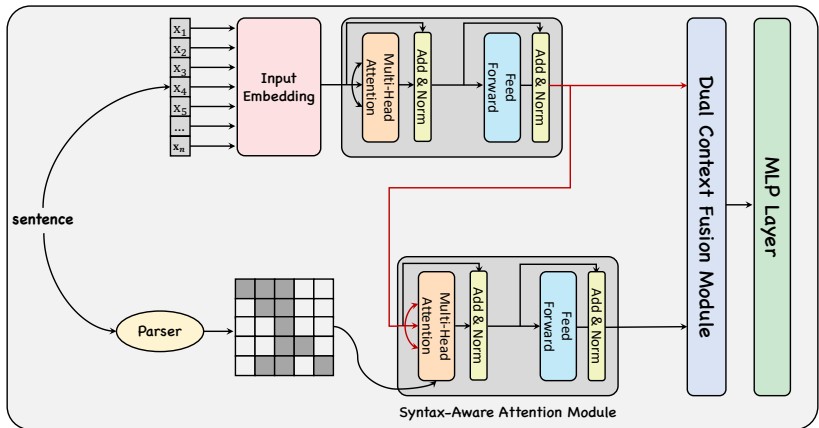

Figure 2: The architecture of SAT. The utterance is first passed to the utterance encoder (§2.1) and SATM (§2.2) to obtain feature representations, which are then fused in DCFM (§2.3) to obtain the final representation (§2.4).

## 2 Methodology

The main architecture mainly consisting of SATM and DCFM is illustrated in Figure 2. We elaborate on each component below.

### 2.1 Backbone Model

Following previous works, we select two backbone models, i.e., BERT (Devlin et al., 2019) and RoBERTa (Liu et al., 2019b). Specifically, for a given utterance $\mathbf{x} = \{x_1, x_2, ..., x_n\}$, we prepend $[CLS]$ to represent the entire sequence and append $[SEP]$ to separate non-consecutive token sequences (Devlin et al., 2019). Subsequently, we obtain the encoded feature representation $\mathbf{H} = \{\mathbf{H}_{CLS}, \mathbf{H}_1, \mathbf{H}_2, ..., \mathbf{H}_n, \mathbf{H}_{SEP}\}$ and feed $\mathbf{H}_{CLS} \in \mathbb{R}^{1 \times d}$ into the backbone model, where $L$ represents the length of the utterance and $d$ is the hidden dimension. Finally, $\mathbf{H}_{CLS}$ is passed through a simple multilayer perceptron (MLP) neural network to predict the intent.

### 2.2 Syntax-Aware Attention Module (SATM)

Unlike previous approaches that treat additional syntactic features as a separate part of PLMs (Strubell et al., 2018; Kasai et al., 2019; Duan et al., 2019), we implement SATM by applying an attention mask on self-attention to enforce syntactic constraints. Specifically, we begin by deriving the syntactic structure from dependency parsing, treating it as a directed tree. Each token $\mathbf{h}_i$ is then mapped to a corresponding tree node $\mathbf{q}_i$. For a given node $\mathbf{q}_i$, we define the set of its syntactic ancestors as $\mathbf{P}_i$. To focus attention on the node and its syntactic ancestors, we obtain a sequence

of SAT mask $\mathcal{M} \in \mathbb{R}^{n*n}$ from the parser:

$$\mathcal{M}[i,j] = \begin{cases} 1, & \mathbf{q}_j \in \mathbf{P}_i \ or \ i = j \\ 0, & otherwise \end{cases} \quad (1)$$

From the formulation, if $\mathcal{M}[i,j] = 1$, it means that the node $\mathbf{q}_j$ is the ancestor of $\mathbf{q}_i$ or $\mathbf{q}_i$ itself.

Afterward, we project the outputs of the backbone model $\mathbf{H}_{CLS}$ into separate query $\mathbf{Q}$, key $\mathbf{K}$, and value $\mathbf{V}$ representations. The syntax-aware attention scores can be calculated as follows:

$$\mathbf{W}_{\text{SATM}} = \text{softmax}(\frac{\mathcal{M} \cdot (\mathbf{QK}^{\top})}{\sqrt{d_k}})\mathbf{V} \quad (2)$$

where $d_k$ is the dimension of key $\mathbf{K}$. Concatenating all heads $\mathbf{W}_{\text{SATM}}$ and passing them through a feed-forward layer, we obtain the syntax-aware attention output $\mathbf{H}_{\text{SATM}}$.

### 2.3 Dual Context Fusion Module (DCFM)

We obtain original and syntax-aware feature representations $\mathbf{H}_{CLS}$ and $\mathbf{H}_{\text{SATM}}$ to enhance the model's performance. Different fusion methods have varying impacts on the representations, motivating us to explore diverse techniques to improve effectiveness. We then explore several representative fusion methods, including concatenation and more complex mechanisms that consider the correlation between the two outputs. The fusion process of two embeddings is defined as: $\mathbf{Y} = \mathbf{H}_{CLS} \oplus \mathbf{H}_{\text{SATM}}$ where $\oplus$ represents fusion method, $\mathbf{H}$ is the final representation for intent detection. Specially, We choose three typical works (Wang et al., 2018; Liu et al., 2020; Li et al., 2019) to compare their effectiveness: (1) LEAM is a way that extracts the weights of features to assign

as a coefficient, (2) aimNet is a symmetrical multi-modal attention module, (3) SKnet is a gate-based fusion method applied to images.

We further propose a **D**ual **C**ontext **F**usion **M**odule (DCFM). Firstly, we introduce $gate_{base}$ that involves extracting useful features from both the original embedding and the syntax embedding. The data is then mapped to a range of 0 to 1, serving as a filter gate. This gate selectively filters the information from the syntax embedding, mitigating the errors caused by ASR and parser to some extent. Subsequently, the filtered data undergoes additional processing using an activation function and a fully connected layer to retain only the relevant and useful features:

$$gate_{base} = \sigma(\mathbf{W}_1[\mathbf{H}_{CLS}, \mathbf{H}_{SATM}] + \mathbf{b}_1) \quad (3)$$

$$\mathbf{H}_a = \mathbf{W}_3(\text{Tanh}(\mathbf{W}_2(gate_{base}\cdot$$
$$\mathbf{H}_{SATM}) + \mathbf{b}_2)) + \mathbf{b}_3 \quad (4)$$

where $[\cdot, \cdot]$ denotes concatenation operation, $\mathbf{W}_*$ and $\mathbf{b}_*$ are trainable weights, $\sigma$ is sigmoid activation (Han and Moraga, 1995) and $Tanh$ is Tanh activation (Kalman and Kwasny, 1992). To ensure the robustness of the model's performance, it is necessary to combine it with the original embedding. Inspired by Self-Adaptive Scaling Approach (Liu et al., 2019a), we utilize a control gate that automatically learns the most relevant and useful features to obtain the final embedding in a robust manner.

$$\mathbf{H}_b = \text{LN}(\mathbf{H}_{CLS} + \mathbf{H}_a) \quad (5)$$

$$\mathbf{Y} = \alpha \cdot \mathbf{H}_{CLS} + \beta \cdot \mathbf{H}_a$$
$$+ (1 - \alpha)(1 - \beta) \cdot \mathbf{H}_b \quad (6)$$

$$1 = \alpha + \beta \quad (7)$$

where $\alpha$ and $\beta$ are hyperparameters and learn by training, and LN stands for layer normalization. $\mathbf{Y}$ remains the ultimate outcome.

## 2.4 Training Objective

Following Cheng et al. (2023a,b), we employ a fully connected network as a task classifier, and optimize SAT using Cross-Entropy.

## 3 Experiment

### 3.1 Datasets and Experimental Settings

We conduct experiments on three widely used benchmark datasets, namely TREC6, TREC50 and ATIS (Hovy et al., 2001; Li and Roth, 2002; Shivakumar et al., 2019). Detail analysis and experimental settings can be seen in Appendix A.

## 3.2 Main Results

Table 1 presents the main results on three benchmark datasets. SAT has shown significant improvements across various performance metrics. For example, on the TREC6 dataset, SAT demonstrates a notable enhancement of 0.6% in accuracy and 0.8% in F1-score compared to the gate unit based on RoBERTa. Thus, we analyze the factors contributing to the success of SAT, which are elaborated as follows:

**(1)** Incorporating syntactic information improved results by capturing keywords with dependency relationships. The utilization of syntactic information proves to be beneficial, particularly in cases where such information is lacking. These findings align with previous studies (Sachan et al., 2021).

**(2)** Considering the non-robust nature of syntax embedding, we explore various fusion methods, finding that the gated unit or gate-based fusion method outperforms other fusion methods. We attribute this effectiveness to SAT's ability to effectively leverage two distinct features, leading to enhanced information utilization, more reasonable weight allocation, and improved modeling of syntactic relations.

**(3)** Errors in ASR and parsing can distort or render token relationships irrelevant. For instance, when the verb "is" is the keyword in an utterance, its relationship may not be the most crucial one. One of the notable advantages of SAT is its ability to mitigate such errors to a certain extent.

## 3.3 Integration with Other Models

We compare SAT with previous works (Sundararaman et al., 2021; Chang and Chen, 2022) on the same benchmark datasets. In Table 2, we find that our proposal SAT outperforms the other models. Moreover, one of the key advantages of SAT is its seamless integration with BERT-based models. This integration has significantly enhanced the overall performance, highlighting the versatility and potential impact of our proposed model in driving major advancements in the field.

## 3.4 Ablation Study on Parser

We also extract syntax information in the form of dependency trees using a high-precision parser, such as the HPSG parser (Zhou and Zhao, 2019), coupled with two different pre-trained backbone models: XL-Net (Yang et al., 2019) and BERT (Devlin et al., 2019). Notably, the HPSG parser with

| Pre-trained Model | Syntax Information | Fusion Module | Fusion Method | Dataset: TREC6 | | | | Dataset: TREC50 | | | | Dataset: ATIS | | | |
|---|---|---|---|---|---|---|---|---|---|---|---|---|---|---|---|
| | | | | Acc | F1 | P | R | Acc | F1 | P | R | Acc | F1 | P | R |
| BERT | ✗ | ✗ | - | 83.9 | 81.1 | 79.9 | 82.6 | 75.8 | 57.8 | 62.5 | 58.9 | 94.4 | 77.6 | 75.2 | 86.8 |
| | ✓ | ✗ | - | 82.8 | 80.1 | 78.4 | 82.7 | 76.2 | 58.3 | 61.2 | 58.6 | 94.6 | 79.4 | 77.2 | 85.9 |
| | ✓ | ✓ | add | 84.4 | 82.3 | 82.3 | 83.4 | 75.2 | 59.1 | 60.9 | 58.3 | 94.6 | 77.8 | 76.4 | 86.3 |
| | ✓ | ✓ | LEAM | 84.2 | 82.4 | 81.5 | 83.7 | 75.9 | 57.5 | 59.7 | 60.1 | 94.9 | 80.1 | 81.2 | 86.8 |
| | ✓ | ✓ | aimNet | 84.8 | 82.3 | 81.2 | 83.8 | 77.0 | 58.5 | 58.0 | 58.7 | 94.8 | 79.6 | 76.5 | 87.2 |
| | ✓ | ✓ | SKnet | 85.2 | 82.5 | 81.4 | 83.9 | 78.0 | 60.8 | 62.1 | 63.4 | 95.1 | 79.8 | 76.3 | 87.4 |
| | ✓ | ✓ | gate unit | 85.6 | 83.8 | 83.8 | 84.8 | 78.2 | 61.8 | 64.2 | **64.6** | 95.2 | 80.8 | 77.8 | 88.5 |
| | ✓ | ✓ | SAT | **86.2** | **85.6** | **86.5** | **85.0** | **78.9** | **62.6** | **65.0** | 64.6 | **95.3** | **82.7** | **81.5** | **88.8** |
| RoBERTa | ✗ | ✗ | - | 84.1 | 82.1 | 82.5 | 83.4 | 76.0 | 59.9 | 60.1 | 61.5 | 94.5 | 79.8 | 77.8 | 86.3 |
| | ✓ | ✗ | - | 83.2 | 81.5 | 79.1 | 82.7 | 76.4 | 60.1 | 62.2 | 64.0 | 94.8 | 81.5 | 79.2 | 87.0 |
| | ✓ | ✓ | add | 84.6 | 84.3 | 84.9 | 83.9 | 75.4 | 61.1 | 62.9 | 63.7 | 94.9 | 81.5 | 78.8 | 87.9 |
| | ✓ | ✓ | LEAM | 84.6 | 83.5 | 83.1 | 84.1 | 76.6 | 57.8 | 62.4 | 60.0 | 95.1 | 81.2 | 78.5 | 87.8 |
| | ✓ | ✓ | aimNet | 85.2 | 83.9 | 83.7 | 84.4 | 77.2 | 59.8 | 61.1 | 63.7 | 95.0 | 80.7 | 78.0 | 87.2 |
| | ✓ | ✓ | SKnet | 85.4 | 84.0 | 84.2 | 83.9 | 78.6 | 60.6 | 64.1 | 61.6 | 95.2 | 80.1 | 76.3 | 88.0 |
| | ✓ | ✓ | gate unit | 85.8 | 85.0 | 86.3 | 84.5 | 78.2 | 62.1 | 64.7 | 64.5 | 95.3 | 81.8 | 81.0 | 88.4 |
| | ✓ | ✓ | SAT | **86.4** | **85.8** | **87.0** | **85.0** | **79.0** | **62.8** | **65.2** | 64.6 | **95.4** | **83.0** | **81.2** | **89.5** |

Table 1: Results on three datasets. "Syntax Information" refers to the inclusion of SAT mask in the backbone model.

| Method | Dataset | | |
|---|---|---|---|
| | TREC6 | TREC50 | ATIS |
| RoBERTa (Liu et al., 2019b) | 84.1 | 76.6 | 94.4 |
| Phoneme-BERT (Sundararaman et al., 2021) | 85.9 | 77.8 | 94.8 |
| SpokenCSE (Chang and Chen, 2022) | 86.4 | 76.2 | 95.1 |
| **Proposed (SAT)** | **86.4** | **79.0** | **95.4** |
| Phoneme-BERT ( + SAT) | 86.6 | **79.4** | 95.6 |
| SpokenCSE (finetune + SAT) | **87.2** | 76.6 | **95.8** |

Table 2: Comparing results of accuracy on benchmarks.

| Fusion Method | Metric: Accuracy | | Δ |
|---|---|---|---|
| | HPSG-Parser-BERT | HPSG-Parser-XL-Net | |
| addition | 84.4 | **84.6** | 0.2 |
| LEAM (Wang et al., 2018) | 84.5 | **84.6** | 0.1 |
| aimNet (Liu et al., 2020) | 85.0 | **85.2** | 0.2 |
| SKnet (Li et al., 2019) | 84.8 | **85.2** | 0.4 |
| gate unit (Li et al., 2018) | 85.2 | **85.6** | 0.4 |
| **DCFM** | 86.3 | **86.4** | 0.1 |

Table 3: Accuracy on TREC6 about using different parser based on RoBERTa.

XL-Net outperforms the HPSG parser with BERT in terms of performance. The results presented in Table 3 clearly demonstrate that a highly accurate parser yields superior outcomes. In other words, there is a direct correlation between the accuracy of the parser and the performance achieved. Furthermore, these findings further emphasize the importance of incorporating syntax dependency information to enhance the intent detection task.

### 3.5 Visualization

In Figure 3, we compare the syntax attention and the ordinary attention using a heat-map of attention scores. The heat-map excludes [CLS] and [SEP] tokens to establish a clearer correlation among the other tokens. We observe that the SAT exhibits

the capacity to recognize crucial information. For example, the keyword "describe" shows high attention scores with the tokens "term" and "word", highlighting SAT's ability to identify relevant features that might be overlooked by the original attention. This finding provides a compelling explanation for the effectiveness of SAT.

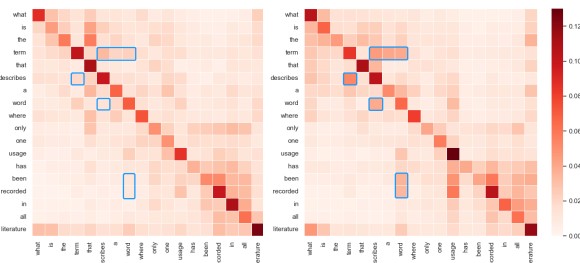

Figure 3: Visualization of attention score selected from last layer and averaged over all heads. **Left** is the original attention, while **right** is the syntax attention.

## 4 Conclusion

In this paper, we propose a novel **S**yntax-**a**ware a**t**tention (SAT) to improve SLU by incorporating syntactic information, specifically dependency structures, into PLM. Experimental results demonstrate the superiority of our method over the backbone BERT/RoBERTa models on three benchmark datasets. Moreover, we experiment with multiple parsers and find that parsers with stronger syntactic parsing abilities lead to more substantial improvements in SLU systems. Detailed analysis confirms the effectiveness and power of SAT in enhancing SLU systems.

## Limitations

While the Syntax-aware attention (SAT) exhibits significant improvements by incorporating syntactic information, it still possesses certain drawbacks. Firstly, our method can only be applied in Transformer architectures that rely on self-attention mechanisms. Without the attention mechanism as a foundation, the utilization of our method would be infeasible. Secondly, SATM hinges on the grammatical relation of "all ancestor nodes". We believe that exploring the relationships among "all ancestor nodes + limited child nodes" could yield valuable insights that deserve further investigation. Thirdly, as a gated mechanism, the introduction of additional parameters by DCFM may potentially increase the risk of overfitting and further complicate the clear explication of its function within the framework.

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

## A Experiments

### A.1 Datasets

We conduct experiments on three widely used benchmark datasets, namely TREC6, TREC50 and ATIS (Hovy et al., 2001; Li and Roth, 2002; Shivakumar et al., 2019; Hemphill et al., 1990). The statistics of these datasets are shown in Table 4. Noted that we employ the original text to generate SAT mask. For training, we utilize the ASR transcript of the speech version of the original texts, following the format and partitioning scheme described in (Sundararaman et al., 2021).

| Dataset | #Class | Avg. Length | Train | Test |
|---------|--------|-------------|-------|------|
| TREC6   | 6      | 8.89        | 5,452 | 500  |
| TREC50  | 50     | 8.89        | 5,452 | 500  |
| ATIS    | 22     | 11.14       | 4,978 | 893  |

Table 4: Data statistics on TREC6, TREC50 and ATIS.

### A.2 Experimental Settings

Considering the inference speed, the following work utilizes the BERT$_{base}$ (Devlin et al., 2019) and RoBERTa$_{base}$ (Liu et al., 2019b) as the backbone models. As for the parser, we employ Head-Driven Phrase Structure Grammar (HPSG) (Zhou and Zhao, 2019) for the syntax analysis of the input utterances. During training, we explore different batch sizes from a range of 32, 64 and 128, and conduct a total of 10 epochs to ensure the convergence. Furthermore, we tune the hyperparameter $\alpha$ and $\beta$ of the gated unit, which are initially set to 0.5, and dynamically adjust them during the training process to achieve optimal performance.

To evaluate the model's effectiveness, we employ accuracy, F1-score, precision, and recall as evaluation metrics.

## B Ablation Study on Different Components

To further verify effectiveness of each component in SAT, we conduct a set of ablation experiments on the TREC6 dataset. As demonstrated in the Table 5, the proposed module presents considerable efficiency. It is important to highlight that, instead of leaning on the syntactic information, we conducted experiments with a standard attention layer to verify the efficacy of the syntax-aware attention. Consequently, we can observe that an improvement

in each metric and believe that module's effect remains substantial and meaningful.

| Model | Dataset: TREC6 | | | |
|-------|-----|-----|-----|-----|
|       | Acc | F1  | P   | R   |
| *Backbone (BERT)* | 83.9 | 81.1 | 79.9 | 82.6 |
| *+ Attenion* | 84.2 (↑0.3) | 81.3 (↑0.2) | 81.2 (↑1.3) | 82.8 (↑0.2) |
| *+ SATM (Instead of Attention)* | 84.6 (↑0.4) | 83.2 (↑1.9) | 83.6 (↑2.4) | 83.8 (↑1.0) |
| *+ SATM + DCFM* | **86.2 (↑1.6)** | **85.6 (↑2.4)** | **86.5 (↑2.9)** | **85.0 (↑1.2)** |

Table 5: Results of ablation test on the TREC6 dataset.

## C Performance Comparison

To clarify the novelty, our work represents one of the first attempts at incorporating a syntax-aware attention method into spoken language understanding research. Furthermore, we introduce a novel integration technique, DCFM (Dual Context Fusion Module). This is a unique contribution of our work which differentiates our approach from existing ones. In Table 6, we compare SAT with approaches such as other methods that incorporate knowledge attention or syntactic information. The results provide evidence that our proposed SATM (Syntax-Aware Attention Module) and DCFM technique successfully improve the task's performance.

| Model | TREC6 | TREC50 | ATIS |
|-------|-------|--------|------|
| K-SAN (Chen et al., 2016) | 84.6 | 76.4 | 94.2 |
| BERT-SIM (Xia et al., 2021) | 85.1 | 76.9 | 94.6 |
| CapuBERT (Nguyen, 2022) | 85.3 | 77.8 | 94.9 |
| SG-Net (Zhang et al., 2020) | 85.2 | 77.6 | 94.7 |
| Syntax-BERT (Bai et al., 2021) | 85.5 | 78.2 | 95.0 |
| **SAT** | **86.4** | **79.0** | **95.4** |

Table 6: Results of comparison on three datasets.