# OpenReview forum: "Syntax Matters: Towards Spoken Language Understanding via Syntax-Aware Attention"
_EMNLP/2023/Conference — EMNLP 2023 Findings_

### Official Review · Reviewer_ajUT · 2023-08-01

**Soundness:** 3

**Excitement:**

3: Ambivalent: It has merits (e.g., it reports state-of-the-art results, the idea is nice), but there are key weaknesses (e.g., it describes incremental work), and it can significantly benefit from another round of revision. However, I won't object to accepting it if my co-reviewers champion it.

**Missing References:**

[1] Chen et al. 2016 “Knowledge as a Teacher: Knowledge-Guided Structural Attention Networks”
[2] Xia et al. 2021 “Using Prior Knowledge to Guide BERT’s Attention in Semantic Textual Matching Tasks”
[3] Nguyen et al. 2022 “Improving Spoken Language Understanding by Enhancing Text Representation”


**Paper Topic And Main Contributions:**

The paper proposes to introduce a syntax-aware mask to guide the attention of each word in transformers toward their syntactic ancestors from a dependency parsing tree. Syntactic information from a pretrained parser will produce a mask to constrain the attention between words and their corresponding ancestors only. The experiments show the benefit of syntactic information in BERT-based models in intent detection while more baselines and references are suggested to support the novelty and usefulness of the proposed framework.

**Questions For The Authors:**

1. From L77-L80, do you mean we obtain the encoded feature representation from the backbone model and feed [CLS] embedding in SATM module?
2. Have you tried to include the mask in the first backbone model directly?

**Reasons To Accept:**

1. The paper is easy to follow and demonstrates the effectiveness of adopting syntactic information to aid the spoken language understanding tasks.
2. Extensive experiments have shown how different fusion mechanisms and syntactic masks may affect the intent detection performance.

**Reasons To Reject:**

1. The novelty is relatively limited where the benefits of external syntactic information and using it as a mask in attention have been explored in previous works such as [1-3]. The technique is the extensive use of combination of existing approaches. However, the baselines such as knowledge attention or other approaches of syntactic information inclusion are not compared.
2. Some references are lacking to support the novelty and usefulness of the proposed framework.

**Reproducibility:**

5: Could easily reproduce the results.

**Reviewer Confidence:**

4: Quite sure. I tried to check the important points carefully. It's unlikely, though conceivable, that I missed something that should affect my ratings.

---

> ### Author Rebuttal · Authors · 2023-08-29
>
> We extend our heartfelt appreciation for the thorough reviews you have provided. We value your insights and would like to offer detailed responses to your concerns:
>
> __R1:__
> Thank you for your feedback on the perceived limited novelty of our work and the need for a broader baseline comparison. We value your comments and agree that more comprehensive comparisons would strengthen the presentation of our work. To clarify the novelty, we would like to mention that our work represents one of the first attempts at incorporating a syntax-aware attention method into spoken language understanding research. Furthermore, we introduce a novel integration technique, DCFM (Dual Context Fusion Module). This is a unique contribution of our work which differentiates our approach from existing ones. The results presented in the table will provide evidence that our proposed SAT (syntax-aware attention) and DCFM technique successfully improve the task's performance, thereby asserting the effectiveness and benefits of our approach.
> |Method|TREC6|TREC50|ATIS|
> |-|-|-|-|
> |K-SAN [1]|84.6|76.4|94.2|
> |BERT-SIM [2]|85.1|76.9|94.6|
> |CapuBERT [3]|85.3|77.8|94.9|
> |SG-net [4]|85.2  |77.6|94.7|
> |Syntax-BERT [5]|85.5  |78.2|95.0|
> |__SAT__|__86.4__ |__79.0__|__95.4__|
>
> __R2:__
> We appreciate the reviewer's feedback about the need for additional references to support the novelty and usefulness of our proposed framework. In response to this concern, we will include the provided references in the main table and perform analysis and comparison in the revised version. We will also enrich the referencing in our paper by including citations that highlight the foundational works in the field and contextualize our framework's positioning within the existing literature. We appreciate the reviewer's vigilance in evaluating the scholarly rigor of our paper and are committed to addressing this concern in the revised version.
>
> __Q1:__
> Thank you for requesting clarification on the feature representation. We apologize if our initial explanation was unclear. Here, we provide a more detailed description of how we obtain and utilize the encoded feature representation in our proposed method:
> 1. First, we pass the input through the backbone model, which is responsible for encoding the input data into a feature representation.
> 2. From the encoded feature representation obtained in the previous step, we extract the [CLS] embedding, which serves as a summarization of the input's contextual information.
> 3. This [CLS] embedding is then used as input to our proposed Syntax-Aware Transformer Module (SATM).
>
> By following these steps, we ensure a structured and streamlined integration of feature representation in our approach. The SATM leverages the [CLS] embedding, which is a condensed representation of the input's syntactic and semantic information, to make informed decisions and improve model performance.
>
> __Q2:__
> Thank you for raising this question. The integration of the mask in the backbone model can be found directly in the second row of each backbone model in Table 1, on Line 221 of Page 4. To address your concerns and provide a clear understanding of our contribution, we present the results here again.
>
> These results demonstrate the effectiveness of integrating the mask directly into the backbone model and showcase the improvements our proposed framework brings to the attention-based models. By providing these findings, we aim to highlight the feasibility and practicality of our approach and its potential impact on the field.
> |Pre-trained Model|Syntax Information|Fusion Module|TREC6| | | |TREC|50| | |ATIS| | | |
> |-|-|-|-|-|-|-|-|-|-|-|-|-|-|-|
> | | | |Acc|F1|P|R|Acc|F1|P|R|Acc|F1|P|R|
> |BERT|✗|✗|83.9|81.1|79.9|82.6|   75.8|57.8|62.5|58.9|   94.4|77.6 |75.2|86.8|
> |BERT|✓|✗|82.8|80.1|78.4|82.7|   76.2|58.3|61.2|58.6|   94.6|79.4|77.2 |85.9|
> |RoBERTa|✗|✗|84.1|82.1|82.5|83.4|   76.0|59.9|60.1|61.5|   94.5|79.8|77.8 |86.3|
> |RoBERTa|✓|✗|83.2|81.5|79.1|82.7|   76.4|60.1|62.2|64.0|   94.8|81.5|79.2 |87.0|
>
> __References__
>
> [1] Chen et al. 2016 "Knowledge as a Teacher: Knowledge-Guided Structural Attention Networks"
>
> [2] Xia et al. 2021 "Using Prior Knowledge to Guide BERT’s Attention in Semantic Textual Matching Tasks"
>
> [3] Nguyen et al. 2022 "Improving Spoken Language Understanding by Enhancing Text Representation"
>
> [4] Zhang et al. 2019 "SG-Net: Syntax Guided Transformer for Language Representation"
>
> [5] Bai et al. 2021 "Syntax-BERT: Improving Pre-trained Transformers with Syntax Trees"

---

### Official Review · Reviewer_EFty · 2023-08-04

**Soundness:** 3

**Excitement:**

3: Ambivalent: It has merits (e.g., it reports state-of-the-art results, the idea is nice), but there are key weaknesses (e.g., it describes incremental work), and it can significantly benefit from another round of revision. However, I won't object to accepting it if my co-reviewers champion it.

**Paper Topic And Main Contributions:**

The paper introduces Syntax-aware Attention (SAT), to an intent detection model and claim that this can help Spoken Language Understanding (SLU) systems in task-oriented dialogue systems. By incorporating syntactic information and employing various feature fusion methods, the proposed SAT is shown to empirically achieve substantial improvements and outperform state-of-the-art models on benchmark datasets (ATIS, TREC).

**Reasons To Accept:**

* Performance Improvements over Phoneme-BERT : Authors claim that their proposed model achieves an F1-score of 85.8% on the TREC50 dataset, surpassing Phoneme-BERT's score of 76.6%.
* Good ablation studies on different fusion mechanisms and backbones along with the effect of using different syntactic parsers to obtain the H_sat representation.
* Detailed evaluation and analysis on multiple benchmark datasets. (ATIS, TREC)
* An novel attempt to incorporate syntactic information into pre-trained language models to improve intent classification in SLU systems.

**Reasons To Reject:**

* Insufficient Analysis: The paper lacks in-depth analysis and discussions of the model's limitations, potential failure cases, and practical challenges. A thorough examination of these aspects is essential to understand the boundaries and implications of the proposed approach.
* Methodological Weaknesses: The paper does not provide a comprehensive analysis of the choices made in the model design, such as the dual context fusion module. A lack of in-depth analysis may raise questions about the robustness and reliability of the proposed approach.

**Reproducibility:**

3: Could reproduce the results with some difficulty. The settings of parameters are underspecified or subjectively determined; the training/evaluation data are not widely available.

**Reviewer Confidence:**

4: Quite sure. I tried to check the important points carefully. It's unlikely, though conceivable, that I missed something that should affect my ratings.

---

> ### Author Rebuttal · Authors · 2023-08-29
>
> We greatly appreciate the feedback provided by the reviewers, which highlighted the limitations of the model in the paper and the need to conduct a comprehensive analysis.
>
> __R1:__
> Addressing the issue of insufficient analysis, we intend to include a dedicated section on limitations within the chapter. This section will delve deeply into the potential constraints of our proposed model, elucidating scenarios where challenges might arise. It aims to provide a clear comprehension of the operational boundaries within which our approach functions. This augmentation is geared towards ensuring that readers are thoroughly informed about the practical implications and confines of our research.
>
> While the Syntax-aware attention (SAT) exhibits significant improvements by incorporating syntactic information, it still possesses certain drawbacks. Firstly, our method can only be used in Transformer architectures that rely on self-attention mechanisms. Without the attention mechanism as a basis, the utilization of our method would be rendered infeasible. Secondly, SAT hinges on the grammatical relation of "all ancestor nodes". We believe that an exploration into capturing related relationships among "all ancestor nodes + limited child nodes" may provide significant insights worthy of further investigation. Thirdly, serving as a gated mechanism, DCFM's introduction of additional parameters could potentially raise the probability of overfitting and further complicate the clear explication of its function within the framework.
>
> __R2:__
> We value your observations regarding the necessity for a more comprehensive analysis of our design decisions.
>
> Firstly, we plan to extensively elaborate on the methodological analysis. Our aim is to provide an in-depth explanation of the rationale behind this design choice, with the goal of offering a more comprehensive and insightful analysis. This effort is intended not only to address the concerns you have raised but also to bolster the overall credibility of our proposed approach.
>
> Secondly, we will include an ablation study in our methodology. As demonstrated in the table, the proposed module presents considerable efficiency. It's important to highlight that, instead of leaning on the syntactic information, we conducted experiments with a standard attention layer to verify the efficacy of the syntax-aware attention. Consequently, we believe that module's effect remains substantial and meaningful.
> |Model|Acc|F1|P|R|
> |-|-|-|-|-|
> |Backbone (BERT)|83.9|81.1|79.9|82.6|
> |+ Attention|84.0 (↑0.1)|81.3 (↑0.2)|80.8 (↑0.9)|82.8 (↑0.2)|
> |+ SATM (Instead of Attention)|84.6 (↑0.6)|83.2 (↑1.9)|83.6 (↑2.8)|83.8 (↑1.0)|
> |+ SATM + DCFM|86.2 (↑1.6)|85.6 (↑2.4)|86.5 (↑2.9)|85.0 (↑1.2)|
>
> We genuinely value the reviewer's input and believe that these revisions will significantly enhance the quality and comprehensiveness of our paper. We deeply apologize for forgetting to add this section. Thanks to the reviewer for pointing out this problem, we will fix it in the revised version.

---

### Official Review · Reviewer_ifvP · 2023-08-05

**Soundness:** 4

**Excitement:**

3: Ambivalent: It has merits (e.g., it reports state-of-the-art results, the idea is nice), but there are key weaknesses (e.g., it describes incremental work), and it can significantly benefit from another round of revision. However, I won't object to accepting it if my co-reviewers champion it.

**Paper Topic And Main Contributions:**

The paper suggests enhancing SLU performance by incorporating syntax information through parser dependencies.

The authors utilize a pre-trained syntactic dependence parse tree to create a syntax-aware attention mask, which includes all ancestors and the word itself. They further introduce a novel Dual Context Fusion Module to integrate the original transformer with their Syntax-aware Attention Module.

The authors observe performance improvements on three publicly available SLU datasets with their approach.

**Reasons To Accept:**

1. Paper is well written and clearly explains their methodology with mathematical formulation
2. Their approach is flexible and can be integrated with any BERT-based PLM.
3. I liked the ablation analysis with different fusion methods.

**Reasons To Reject:**

1. Novelty seems incremental to me. What are the ways in which this paper differs from https://aclanthology.org/2021.findings-acl.57.pdf? Is it just applying a very similar methodology to new task?
2. Performance gains seem small. There should be p-test or atleast confidence intervals to check statistical significance.


**Reproducibility:**

4: Could mostly reproduce the results, but there may be some variation because of sample variance or minor variations in their interpretation of the protocol or method.

**Reviewer Confidence:**

3: Pretty sure, but there's a chance I missed something. Although I have a good feel for this area in general, I did not carefully check the paper's details, e.g., the math, experimental design, or novelty.

---

> ### Author Rebuttal · Authors · 2023-08-29
>
> We express our gratitude for the insightful reviews you've provided. We value your input and would like to address your concerns as outlined below.
>
> __R1:__
> Thank you for pointing out the reference [1]. We will include this reference in the revised version and highlight the differences between our approach and theirs, as well as compare it with other frameworks that utilize syntactic information. After examining this paper, we noticed that their primary focus was on using a mask with syntax information and combining it with a gated unit to derive the final attention scores. In contrast, our approach doesn't directly combine elements within the attention block. Instead, our method emphasizes the exploration of integrating the original embedding with an embedding that incorporates syntax information, which distinguishes it from prior research.
>
> __R2:__
> Thank you for raising this concern.
> - Firstly, we would like to emphasize that although the gains may seem small, our framework holds the potential to be readily and seamlessly applied to other attention-based models, as demonstrated in the following Table, where the integration of our framework has led to favorable performance improvements.
> |Method|TREC6|TREC50|ATIS|
> |-|-|-|-|
> |RoBERTa [2]|84.1|76.6|94.4|
> |RoBERTa ( + __SAT__)|__86.4__|__79.0__|__95.4__|
> |Phoneme-BERT [3]|85.9|77.8|94.8|
> |Phoneme-BERT ( + __SAT__)|86.6|__79.4__|95.6|
> |SpokenCSE [4]|86.4|76.2|95.1|
> |SpokenCSE (finetune + __SAT__)|__87.2__|76.6|__95.8__|
>
> - Secondly, we acknowledge the importance of rigorously evaluating the statistical significance of these enhancements. To that end, we conduct experiments using various backbones on the TREC6 dataset and employ a t-test (p<0.05) to validate our claims on performance improvements.
> |Backbone|Acc|F1|P|R|
> |-|-|-|-|-|
> |ALBERT [5]|86.3|85.5|86.8|84.7|
> |ELECTRA [6]|86.8|86.0|87.2|85.4|
> |DeBERTaV3 [7]|87.5|86.7|87.8|85.9|
>
> In our revised version, we will include these statistical measures to support the significance of the reported performance improvements.
>
> __References__
>
> [1] Li et al. 2021 "Improving BERT with Syntax-aware Local Attention"
>
> [2] Liu et al. 2019 "RoBERTa: A Robustly Optimized BERT Pretraining Approach"
>
> [3] Sundararaman et al. 2021 "Phoneme-BERT: Joint Language Modelling of Phoneme Sequence and ASR Transcript"
>
> [4] Chang et al. 2022 "Contrastive Learning for Improving ASR Robustness in Spoken Language Understanding"
>
> [5] Lan et al. 2020 "ALBERT: A Lite BERT for Self-supervised Learning of Language Representations"
>
> [6] Clark et al. 2021 "ELECTRA: Pre-training Text Encoders as Discriminators Rather Than Generators"
>
> [7] He et al. 2021 "DeBERTaV3: Improving DeBERTa using ELECTRA-Style Pre-Training with Gradient-Disentangled Embedding Sharing"

---

### Meta-Review · Area_Chair_8Smc · 2023-09-16

**Recommendation:** 3

**Metareview:**

This paper proposes to restrain the attention of a tranformer LM based on syntax. They show that this improves performance across multiple SLU datasets. The approach is easily used in many LMs (as long as they use attention), and the paper is well written. The reviewers mentioned multiple competing approaches that aim to incorporate syntax into LMs; which are not compared to (both descriptive and performance). Furthermore, ablation studies and qualitative analysis are not included; limiting the conclusions we can draw from the current paper (it would probably fit better as a long paper including the missing parts).

PS: I think SLU is a very misleading name; as it is not speech, and it is not fully understanding either. I  would suggest to refer to this as intent detection or intent classification

---

### Decision · Program_Chairs · 2023-10-07

**Decision:**

Accept-Findings

**Comment:**

This paper proposes to restrain the attention of a tranformer LM based on syntax. They show that this improves performance across multiple SLU datasets. The approach is easily used in many LMs (as long as they use attention), and the paper is well written. The reviewers mentioned multiple competing approaches that aim to incorporate syntax into LMs; which are not compared to (both descriptive and performance). Furthermore, ablation studies and qualitative analysis are not included; limiting the conclusions we can draw from the current paper (it would probably fit better as a long paper including the missing parts).

PS: I think SLU is a very misleading name; as it is not speech, and it is not fully understanding either. I  would suggest to refer to this as intent detection or intent classification